# $f$-GAIL: Learning $f$-Divergence for Generative Adversarial Imitation Learning

**Xin Zhang[†], Yanhua Li[†], Ziming Zhang[†], Zhi-Li Zhang[$]**
Worcester Polytechnic Institute, USA[†], University of Minnesota, USA[$]
{xzhang17,yli15,zzhang15}@wpi.edu, zhzhang@cs.umn.edu

## Abstract

Imitation learning (IL) aims to learn a policy from expert demonstrations that minimizes the discrepancy between the learner and expert behaviors. Various imitation learning algorithms have been proposed with different *pre-determined* divergences to quantify the discrepancy. This naturally gives rise to the following question: *Given a set of expert demonstrations, which divergence can recover the expert policy more accurately with higher data efficiency?* In this work, we propose $f$-GAIL, a new generative adversarial imitation learning (GAIL) model, that automatically learns a discrepancy measure from the $f$-divergence family as well as a policy capable of producing expert-like behaviors. Compared with IL baselines with various predefined divergence measures, $f$-GAIL learns better policies with higher data efficiency in six physics-based control tasks.

## 1 Introduction

Imitation Learning (IL) or Learning from Demonstrations (LfD) [1, 6, 18] aims to learn a policy directly from expert demonstrations, without access to the environment for more data or any reward signal. One successful IL paradigm is Generative Adversarial Imitation Learning (GAIL) [18], which employs generative adversarial network (GAN) [15] to jointly learn a generator (as a stochastic policy) to mimic expert behaviors, and a discriminator (as a reward signal) to distinguish the generated vs expert behaviors. The learned policy produces behaviors similar to the expert, and the similarity is evaluated using the reward signal, in Jensen-Shannon (JS) divergence (with a constant shift of $\log 4$ [24]) between the distributions of learner vs expert behaviors. Thus, GAIL can be viewed as a variational divergence minimization (VDM) [25] problem with JS-divergence as the objective.

Beyond JS-divergence (as originally employed in GAIL), variations of GAIL have been proposed [18, 13, 12, 20, 14], essentially using different divergence measures from the $f$-divergence family [24, 25], for example, behavioral cloning (BC) [26] with Kullback–Leibler (KL) divergence [24], AIRL [13] and RKL-VIM [20] with reverse KL (RKL) divergence [24], and DAGGER [28] with the Total Variation (TV) [7]. Choosing the right divergence is crucial in order to recover the expert policy more accurately with high data efficiency (as observed in [20, 14, 18, 13, 25, 33]).

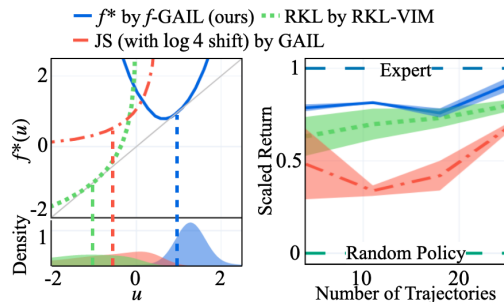

(a) $f^*$ of $f$-divergence. (b) Return of learned policies.

Figure 1: $f$-divergences and policies from GAIL, RKL-VIM, and $f$-GAIL on *Walker* task [32].

**Motivation.** All the above literature works rely on a *fixed* divergence measure manually chosen *a priori* from a set of well-known divergence measures (with an explicit analytic form), e.g., KL, RKL, JS, ignoring the large space of all potential

divergences. Thus, the resulting IL network likely learns a sub-optimal learner policy. For example, Fig. 1 shows the results from GAIL [18] and RKL-VIM [20], which employ JS and RKL divergences, respectively. The learned input density distributions (to the divergence functions) are quite dispersed (thus with large overall divergence) in Fig. 1(a), leading to learner policies with only 30%-70% expert return in Fig. 1(b). In this work, we are motivated to develop a *learnable* model to search and automatically find an appropriate discrepancy measure from the $f$-divergence family for GAIL.

**Our $f$-GAIL.** We propose $f$-GAIL – a new generative adversarial imitation learning model, with a *learnable $f$-divergence* from the underlying expert demonstrations. The model automatically learns an $f$-divergence between expert and learner behaviors, and a policy that produces expert-like behaviors. In particular, we propose a deep neural network structure to model the $f$-divergence space. Fig. 1 shows a quick view of our results: $f$-GAIL learns a new and unique $f$-divergence, with more concentrated input density distribution (thus smaller overall divergence) than JS and RKL in Fig. 1(a); and its learner policy has higher performance (80%-95% expert return) in Fig. 1(b) (See more details in Sec 4). The code for reproducing the experiments are available at `https://github.com/fGAIL3456/fGAIL`. Our key contributions are summarized below:

- We are the *first* to model imitation learning with a ***learnable divergence measure*** from $f$-divergence space, which yields better learner policies, than pre-defined divergence choices (Sec 2).
- We develop an $f^*$-network structure, to model the space of $f$-divergence family, by enforcing two constraints, including i) convexity and ii) $f(1) = 0$ (Sec 3).
- We present promising comparison results of learned $f$-divergences and the performances of learned policies with baselines in six different physics-based control tasks (Sec 4).

## 2   Problem Definition

### 2.1   Preliminaries

**Markov Decision Processes (MDPs).** In an MDP denoted as a 6-tuple $\langle \mathcal{S}, \mathcal{A}, \mathcal{P}, r, \rho_0, \gamma \rangle$ where $\mathcal{S}$ is a set of states, $\mathcal{A}$ is a set of actions, $\mathcal{P} : \mathcal{S} \times \mathcal{A} \times \mathcal{S} \mapsto [0, 1]$ is the transition probability distribution, $r : \mathcal{S} \times \mathcal{A} \mapsto \mathbb{R}$ is the reward function, $\rho_0 : \mathcal{S} \mapsto \mathbb{R}$ is the distribution of the initial state $s_0$, and $\gamma \in [0, 1]$ is the discount factor. We denote the expert policy as $\pi_E$, and the learner policy as $\pi$. In addition, we use an expectation with respect to a policy $\pi$ to denote an expectation with respect to the trajectories it generates: $\mathbb{E}_\pi[h(s, a)] \triangleq \mathbb{E}[\sum_{t=0}^\infty \gamma^t h(s_t, a_t)]$, with $s_0 \sim \rho_0, a_t \sim \pi(a_t|s_t), s_{t+1} \sim \mathcal{P}(s_{t+1}|s_t, a_t)$ and $h$ as any function.

$f$**-Divergence.** $f$-Divergence [24, 23, 11] is a broad class of divergences that measures the difference between two probability distributions. Different choices of $f$ functions recover different divergences, e.g. the Kullback-Leibler (KL) divergence, Jensen-Shannon (JS) divergence, or total variation (TV) distance [22]. Given two distributions $P$ and $Q$, an absolutely continuous density function $p(x)$ and $q(x)$ over a finite set of random variables $x$ defined on the domain $\mathcal{X}$, an $f$-divergence is defined as

$$D_f(P\|Q) = \int_\mathcal{X} q(x) f\left(\frac{p(x)}{q(x)}\right) \mathrm{d}x, \tag{1}$$

with the generator function $f : \mathbb{R}_+ \to \mathbb{R}$ as a convex, lower-semicontinuous function satisfying $f(1) = 0$. The *convex conjugate* function $f^*$ also known as the *Fenchel conjugate* [16] is $f^*(u) = \sup_{v \in \mathrm{dom}_f} \{vu - f(v)\}$. $D_f(P\|Q)$ is lower bounded by its variational transformation, i.e., $D_f(P\|Q) \geq \sup_{u \in \mathrm{dom}_{f^*}} \{\mathbb{E}_{x \sim P}[u] - \mathbb{E}_{x \sim Q}[f^*(u)]\}$ (See more details in [25]). Common choices of $f$ functions are summarized in Tab. 1 and the plots of corresponding $f^*$ are visualized in Fig. 4.

**Imitation Learning as Variational $f$-Divergence Minimization (VDM).** Imitation learning aims to learn a policy for performing a task directly from expert demonstrations. GAIL [18] is an IL solution employing GAN [15] structure, that jointly learns a generator (i.e., learner policy) and a discriminator (i.e., reward signal). In the training process of GAIL, the learner policy imitates the behaviors from the expert policy $\pi_E$, to match the generated state-action distribution with that of the expert. The distance between these two distributions, measured by JS divergence, is minimized. Thus the GAIL objective is stated as follows:

$$\min_\pi \max_T \mathbb{E}_{\pi_E}[\log T(s, a)] + \mathbb{E}_\pi[\log(1 - T(s, a))] - \mathcal{H}(\pi), \tag{2}$$

where $T$ is a binary classifier distinguishing state-action pairs generated by $\pi$ vs $\pi_E$, and it can be viewed as a reward signal used to guide the training of policy $\pi$. $\mathcal{H}(\pi) = \mathbb{E}_\pi[-\log \pi(a|s)]$ is the $\gamma$-discounted causal entropy of the policy $\pi$ [18]. Using the variational lower bound of an $f$-divergence, several studies [20, 14, 25, 5] have extended GAIL to a general variational $f$-divergence minimization (VDM) problem for a fixed $f$-divergence (defined by a generator function $f$), with an objective below,

$$\min_\pi \max_T \mathbb{E}_{\pi_E}[T(s,a)] - \mathbb{E}_\pi[f^*(T(s,a))] - \mathcal{H}(\pi). \tag{3}$$

However, all these works rely on manually choosing an $f$-divergence measure, i.e., $f^*$, which is limited by those well-known $f$-divergence choices (ignoring the large space of all potential $f$-divergences), thus lead to a sub-optimal learner policy. Hence, we are motivated to develop a new and more general GAIL model, which automatically searches an $f$-divergence from the $f$-divergence space given expert demonstrations.

## 2.2 Problem Definition: Imitation Learning with Learnable $f$-Divergence.

**Divergence Choice Matters!** As observed in [20, 14, 13, 25, 33], given an imitation learning task, defined by a set of expert demonstrations, different divergence choices lead to different learner policies. Taking KL divergence and RKL divergence (defined in eq. (4) below) as an example, let $p(x)$ be the true distribution, and $q(x)$ be the approximate distribution learned by minimizing its divergence from $p(x)$. With KL divergence, the difference between $p(x)$ and $q(x)$ is weighted by $p(x)$. Thus, in the ranges of $x$ with $p(x) = 0$, the discrepancy of $q(x) > 0$ from $p(x)$ will be ignored. On the other hand, with RKL divergence, $q(x)$ becomes the weight. In the ranges of $x$ with $q(x) = 0$, RKL divergence does not capture the discrepancy of $q(x)$ from $p(x) > 0$. Hence, KL divergence can be used to better learn multiple modes from a true distribution $p(x)$ (i.e., for mode-covering), while RKL divergence will perform better in learning a single mode (i.e., for mode-seeking).

$$D_{KL}(P\|Q) = \int_\mathcal{X} p(x) \log\left(\frac{p(x)}{q(x)}\right) dx, \qquad D_{RKL}(P\|Q) = \int_\mathcal{X} q(x) \log\left(\frac{q(x)}{p(x)}\right) dx. \tag{4}$$

Beyond KL and RKL divergences, there are infinitely many choices in the $f$-divergence family, where each divergence measures the discrepancy between expert vs learner distributions from a unique perspective. Hence, choosing the right divergence for an imitation learning task is crucial and can more accurately recover the expert policy with higher data efficiency.

$f$-**GAIL: Imitation Learning with Learnable $f$-Divergence.** Given a set of expert demonstrations to imitate and learn from, the $f$-divergence, that can highly evaluate the discrepancy between the learner and expert distributions (i.e., the largest $f$-divergence from the family), can better guide the learner to learn from the expert (as having larger improvement margin). As a result, in addition to the policy function $\pi$, the reward signal function $T$, we aim to learn a (convex conjugate) generator function $f^*$ as a regularization term to the objective. The $f$-GAIL objective is as follows,

$$\min_\pi \max_{f^* \in \mathcal{F}^*, T} \mathbb{E}_{\pi_E}[T(s,a)] - \mathbb{E}_\pi[f^*(T(s,a))] - \mathcal{H}(\pi), \tag{5}$$

where $\mathcal{F}^*$ denotes the admissible function space of $f^*$, namely, each function in $\mathcal{F}^*$ represents a valid $f$-divergence. The conditions for a generator function $f$ to represent an $f$-divergence include: i) convexity and ii) $f(1) = 0$. In other words, the corresponding convex conjugate $f^*$ needs to be i) convex (**the convexity constraint**), ii) $\inf_{u \in \mathrm{dom}_{f^*}}\{f^*(u) - u\} = 0$ (**the zero gap constraint**, namely, the minimum distance from $f^*(u)$ to $u$ is 0)[1]. Functions satisfying these two conditions form the admissible space $\mathcal{F}^*$. Note that the zero gap constraint can be obtained by combining convex conjugate $f(v) = \sup_{u \in \mathrm{dom}_{f^*}}\{uv - f^*(u)\}$ and $f(1) = 0$. Tab. 1[2] below shows a comparison of our proposed $f$-GAIL with the state-of-the-art GAIL models [18, 13, 14, 20]. These models use pre-defined $f$-divergences, where $f$-GAIL can learn an $f$-divergence from $f$-divergence family.

Table 1: $f$-Divergence and imitation learning (JS$^*$ is a constant shift of JS divergence by $\log 4$).

| Divergence | KL | RKL | JS$^*$ | *Learned $f$-div.* |
|---|---|---|---|---|
| $f^*(u)$ | $e^{u-1}$ | $-1 - \log(-u)$ | $-\log(1 - e^u)$ | $f^* \in \mathcal{F}^*$ from eq. (5) |
| IL Method | FAIRL[14] | RKL-VIM[20], AIRL[13] | GAIL[18] | $f$-GAIL (Ours) |

[1]Convex and zero-gap constraints are necessary and sufficient conditions to guarantee an $f$-divergence, based on $f^{**} = f$ (see §3.3.2 in [9]) for convex functions, i.e., $f(1) = f^{**}(1) = \max_u\{u - f^*(u)\} = 0$.

[2]Similar observations can be found in [20, 14].

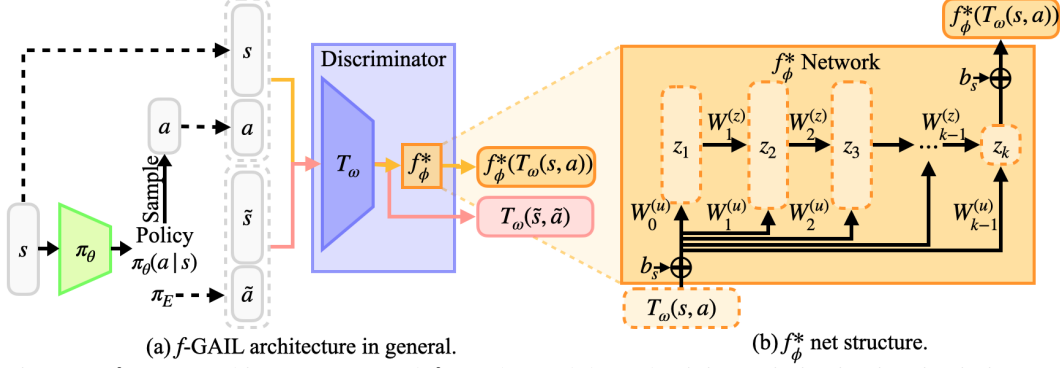

(a) $f$-GAIL architecture in general.　　　　　　　　　(b) $f_\phi^*$ net structure.

Figure 2: $f$-GAIL architecture. ($T_\omega$ and $f_\phi^*$ are learned through a joint optimization in Discriminator.)

# 3 Imitation Learning with Learnable $f$-Divergence

There are three functions to learn in the $f$-GAIL objective in eq. (5), including the policy $\pi$, the $f^*$-function $f^*$, and the reward signal $T$, where we model them with three deep neural networks parameterized by $\theta, \omega$ and $\phi$ respectively. Following the generative-adversarial approach [15], $f_\phi^*$ and $T_\omega$ networks together can be viewed as a discriminator. The policy network $\pi_\theta$ is the generator. As a result, the goal is to find the saddle-point of the objective in eq. (5), where we minimize it with respect to $\theta$ and maximize it with respect to $\omega$ and $\phi$. In this section, we will tackle two key challenges including i) how to design an algorithm to jointly learn all three networks to solve the $f$-GAIL problem in eq. (5)? (See Sec 3.1); and ii) how to design the $f_\phi^*$ network structure to enforce it to represent a valid $f$-divergence? (See Sec 3.2). Fig. 2 shows the overall $f$-GAIL model structure.

## 3.1 $f$-GAIL Algorithm

Our proposed $f$-GAIL algorithm is presented in Alg. 1. It uses the alternating gradient method (instead of one-step gradient method in $f$-GAN [25]) to first update the $f^*$-function $f_\phi^*$ and the reward signal $T_\omega$ in a single back-propagation, and then update the policy $\pi_\theta$. It utilizes Adam [21] gradient step on $\omega$ to increase the objective in eq. (5) with respect to both $T_\omega$ and $f_\phi^*$, followed by a shifting operation on $f_\phi^*$ to guarantee the zero gap constraint (See Sec 3.2 and eq. (7)). Then, it uses the Trust Region Policy Optimization (TRPO) [29] step on $\theta$ to decrease eq. (7) with respect to $\pi_\theta$.

---

**Algorithm 1** $f$-GAIL

---

**Require:** Initialize parameters of policy $\pi_\theta$, reward signal $T_\omega$, and $f_\phi^*$ networks as $\theta_0$, $\omega_0$ and $\phi_0$ (with shifting operation eq. (7) required on $\phi_0$ to enforce the zero gap constraint); expert trajectories $\tau_E \sim \pi_E$ containing state-action pairs.
**Ensure:** Learned policy $\pi_\theta$, $f^*$-function $f_\phi^*$ and reward signal $T_\omega$.
1: **for** each epoch $i = 0, 1, 2, ...$ **do**
2: 　　Sample trajectories $\tau_i \sim \pi_{\theta_i}$.
3: 　　Sample state-action pairs: $\mathcal{D}_E \sim \tau_E$ and $\mathcal{D}_i \sim \tau_i$ with the same batch size.
4: 　　Update $\omega_i$ to $\omega_{i+1}$ and $\phi_i$ to $\phi_{i+1}$ by ascending with the gradients:
$$\Delta_{w_i} = \hat{\mathbb{E}}_{\mathcal{D}_E}[\nabla_{\omega_i} T_{\omega_i}(s,a)] - \hat{\mathbb{E}}_{\mathcal{D}_i}[\nabla_{\omega_i} f_{\phi_i}^*(T_{\omega_i}(s,a))], \quad \Delta_{\phi_i} = -\hat{\mathbb{E}}_{\mathcal{D}_i}[\nabla_{\phi_i} f_{\phi_i}^*(T_{\omega_i}(s,a))].$$
5: 　　Estimate the minimum gap $\delta$ with gradient descent in Alg. 2 and shift $f_{\phi_{i+1}}^*$ (by eq. 7).
6: 　　Take a policy step from $\theta_i$ to $\theta_{i+1}$, using the TRPO update rule to decrease the objective:
$$-\hat{\mathbb{E}}_{\mathcal{D}_i}[f_{\phi_{i+1}}^*(T_{\omega_{i+1}}(s,a))] - \mathcal{H}(\pi_{\theta_i}).$$
7: **end for**

---

## 3.2 Enforcing $f_\phi^*$ Network to Represent the $f$-Divergence Space

The architecture of the $f_\phi^*$ network is crucial to obtain a family of convex conjugate generator functions $f^*$ that represents the entire $f$-divergence space. To achieve this goal, two constraints need to be guaranteed (as discussed in Sec 3.2), including i) the **convexity constraint**, i.e., $f^*(u)$ is convex, and ii) the **zero gap constraint**, i.e., $\inf_{u \in \text{dom}_{f^*}} \{f^*(u) - u\} = 0$. To enforce the convex constraint,

we implement the $f_\phi^*$ network with a neural network structure convex to its input. Moreover, in each epoch, we estimate the minimum gap of $\delta = \inf_{u \in \text{dom}_{f^*}} \{f^*(u) - u\}$, with which we shift it to enforce the zero gap constraint. Below, we detail the design of the $f_\phi^*$ network.

**1. Convexity constraint on $f_\phi^*$ network.** The $f_\phi^*$ network takes a scalar input $u$ from the reward signal network $T_\omega$ output, i.e., $u = T_\omega(s, a)$, with $(s, a)$ as a state-action pair generated by $\pi_\theta$. To ensure the convexity of the $f_\phi^*$ network, we employ the structure of a fully input convex neural network (FICNN) [3] with a composition of convex nonlinearites (e.g., ReLU) and linear mappings (See Fig. 2). The convex structure consists of multiple layer perceptrons. Differing from a fully connected feedforward structure, it includes shortcuts from the input layer $u$ to all subsequent layers, i.e., for each layer $i = 0, \cdots, k - 1$,

$$z_{i+1} = g_i(W_i^{(z)} z_i + W_i^{(u)} z_0 + b_i), \quad \text{with} \quad f_\phi^*(u) = z_k + b_s \quad \text{and} \quad z_0 = u + b_s, \quad (6)$$

where $z_i$ denotes the $i$-th layer activation, $g_i$ represents non-linear activation functions, with $W_0^{(z)} \equiv 0$. $b_s$ is a bias over both the input $u$ and the last layer output $z_k$, which is used to enforce the zero gap constraint (as detailed below). As a result, the parameters in $f_\phi^*$ include $\phi = \{W_{0:k-1}^{(u)}, W_{1:k-1}^{(z)}, b_{0:k-1}, b_s\}$. Restricting $W_{1:k-1}^{(z)}$ to be non-negative and $g_i$'s to be convex non-decreasing activation functions (e.g. ReLU) guarantee the network output to be convex to the input $u = T_\omega(s, a)$. The convexity follows the fact that a non-negative sum of convex functions is convex and that the composition of a convex and convex non-decreasing function is also convex [9]. To ensure the non-negativity on $W_{1:k-1}^{(z)}$, in the training process, we clip the $W_{1:k-1}^{(z)}$ to be at least 0, i.e., $w = \max\{0, w\}$ for $\forall w \in W_{1:k-1}^{(z)}$, after each update to $\phi$.

**2. Zero gap constraint on $f_\phi^*$ network**, i.e., $\inf_{u \in \text{dom}_{f_\phi^*}} \{f_\phi^*(u) - u\} = 0$. This constraint requires $f_\phi^*(u) \geq u$ for $\forall u \in \text{dom}_{f^*}$, with the equality attained. For a general convex function $f_\phi^*(u)$, its gap from $u$, defined as $\delta = \inf_{u \in \text{dom}_{f_\phi^*}} \{f_\phi^*(u) - u\}$, is not necessarily zero. We enforce the zero gap constraint by estimating $\delta$ and shifting $f_\phi^*(u)$ based on $\delta$ in each training epoch. We directly estimate the minimum gap $\delta$ by gradient descent with respect to $u$. Using $\delta$, we shift $f_\phi^*(u)$ as follows,

$$f_{\phi'}^*(u) = f_\phi^*\left(u - \frac{\delta}{2}\right) - \frac{\delta}{2}, \text{where } \delta = \inf_{u \in \text{dom}_{f_\phi^*}} \{f_\phi^*(u) - u\}. \quad (7)$$

This shift guarantees zero gap constraint, and we delegate the proof to Appendix A. In each epoch, the estimation process of $\delta$ is detailed in Alg. 2, and the shift operation is implemented by updating $b_s' = b_s - \delta/2$. Fig. 3 illustrates the operations of estimating $\delta$ and shifting $f_\phi^*$. Note that $\delta$ represents the minimum gap in function value between $f_\phi^*(u)$ and $u$. Shifting $\delta/2$ over both input and output space of $f_\phi^*(u)$ (i.e., Line 5 in Alg. 1) enforces the zero gap constraint. Note that this shifting operation is also performed, when initializing the parameters $\phi_0$ for $f_\phi^*(u)$, to make sure the training starts from a valid $f$-divergence[3].

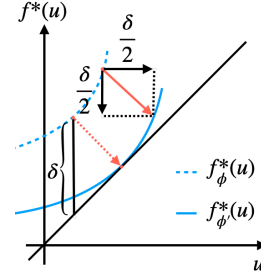

Figure 3: Illustration of shifting $f_\phi^*$.

**Algorithm 2** $\delta$ Estimation

**Require:** $f_\phi^*$ network; initial $u_0$; $\eta > 0$.
**Ensure:** $\delta$.
1: **for** $i = 1, 2, \ldots$ **do**
2: $\quad h = \nabla_u f_\phi^*(u) - 1$;
3: $\quad u_i = u_{i-1} - \eta \cdot h$;
4: **end for**
5: $\delta = f_\phi^*(u_i) - u_i$.

## 4 Experiments

We evaluate Alg. 1 by comparing it with baselines on six physical-based control tasks, including the CartPole [8] from the classic RL literature, and five complex tasks simulated with MuJoCo [32], such as HalfCheetah, Hopper, Reacher, Walker, and Humanoid. By conducting experiments on these tasks,

we show that *i)* our $f$-GAIL algorithm can learn diverse $f$-divergences, comparing to the limited choices in the literature (See Sec 4.1); *ii)* $f$-GAIL algorithm always learn policies performing better than baselines (See Sec 4.2); *iii)* $f$-GAIL algorithm is robust in performance with respect to structure changes in the $f_\phi^*$ network (See Sec 4.3).

Each task in the experiment comes with a true reward function, defined in the OpenAI Gym [10]. We first use these true reward functions to train expert policies with trust region policy optimization (TRPO) [29]. The trained expert policies are then utilized to generate expert demonstrations. To evaluate the data efficiency of $f$-GAIL algorithm, we sampled datasets of varying trajectory counts from the expert policies, while each trajectory consists of about $50$ state-action pairs. Below are five IL baselines, we implemented to compare against $f$-GAIL.

- *Behavior cloning (BC)* [26]: A set of expert state-action pairs is split into 70% training data and 30% validation data. The policy is trained with supervised learning. BC can be viewed as minimizing KL divergence between expert's and learner's policies [20, 14].
- *Generative adversarial imitation learning (GAIL)* [18]: GAIL is an IL method using GAN architecture [15], that minimizes JS divergence between expert's and learner's behavior distributions.
- *BC initialized GAIL (BC+GAIL)*: As discussed in GAIL [18], BC initialized GAIL will help boost GAIL performance. We pre-train a policy with BC and use it as initial parameters to train GAIL.
- *Adversarial inverse reinforcement learning (AIRL)* [13]: AIRL applies the adversarial training approach to recover the reward function and its policy at the same time, which is equivalent to minimizing the reverse KL (RKL) divergence of state-action visitation frequencies between the expert and the learner [14].
- *Reverse KL - variational imitation (RKL-VIM)* [20]: the algorithm uses the RKL divergence instead of the JS divergence to quantify the divergence between expert and learner in GAIL architecture[4].

For fair comparisons, the policy network structures $\pi_\theta$ of all the baselines and $f$-GAIL are the same in all experiments, with two hidden layers of 100 units each, and tanh nonlinearlities in between. The implementations of reward signal networks and discriminators vary according to baseline architectures, and we delegate these implementation details to Appendix B. All networks were always initialized randomly at the start of each trial. For each task, we gave GAIL, BC+GAIL, AIRL, RKL-VIM and $f$-GAIL exactly the same amount of environment interactions for training.

## 4.1 $f_\phi^*$ Learned from $f$-GAIL

Fig. 4 shows that $f$-GAIL learned unique $f_\phi^*(u)$ functions for all six tasks, and they are different from those well-known divergences, such as RKL and JS divergences. Clearly, the learned $f_\phi^*(u)$'s are convex and with zero gap from $u$, thus represent valid $f$-divergences. Moreover, *the learned f-divergences are similar, when the underlying tasks share commonalities.* For example, the two $f_\phi^*(u)$ functions learned from CartPole and Reacher tasks (Fig. 4(a) and (d)) are similar, because the two tasks are similar, i.e., both aiming to keep a balanced distance from the controlling agent to a target. On the other hand, both Hopper and Walker tasks aim to train the agents (with one foot for Hopper and two feet for Walker) to proceed as fast as possible, thus their learned $f_\phi^*(u)$ are similar (Fig. 4(c) and (e)). (See Appendix B for descriptions and screenshots of tasks.) We also plot Fig. 5 to show that given a task, the learned $f^*$ functions are consistent (small variance) for different sample sizes. Similar observations are made for tasks CartPole, Reacher and Humanoid as well.

In state-of-the-art IL approaches and our $f$-GAIL (from eq. (3) and (5)), the $f^*$-function takes the learner reward signal $u = T_\omega(s, a)$ (over generated state-action pairs $(s, a)$'s) as input. By examining the distribution of $u$, two criteria can indicate that the learner policy $\pi_\theta$ is close to the expert $\pi_E$:

i. $u$ centers around zero gap, i.e., $f^*(u) - u \approx 0$. This corresponds to the generator function $f$ centered around $f(p(s, a)/q(s, a)) \approx f(1) = 0$, with $p$ and $q$ as the expert vs learner distributions;
ii. $u$ has small standard deviation. This means that $u$ concentrates on the nearby range of zero gap, leading to a small $f$-divergence between learner and expert, since $D_f(p(s, a)\|q(s, a)) \approx \int q(s, a)f(1)\mathrm{d}(s, a) = 0$.

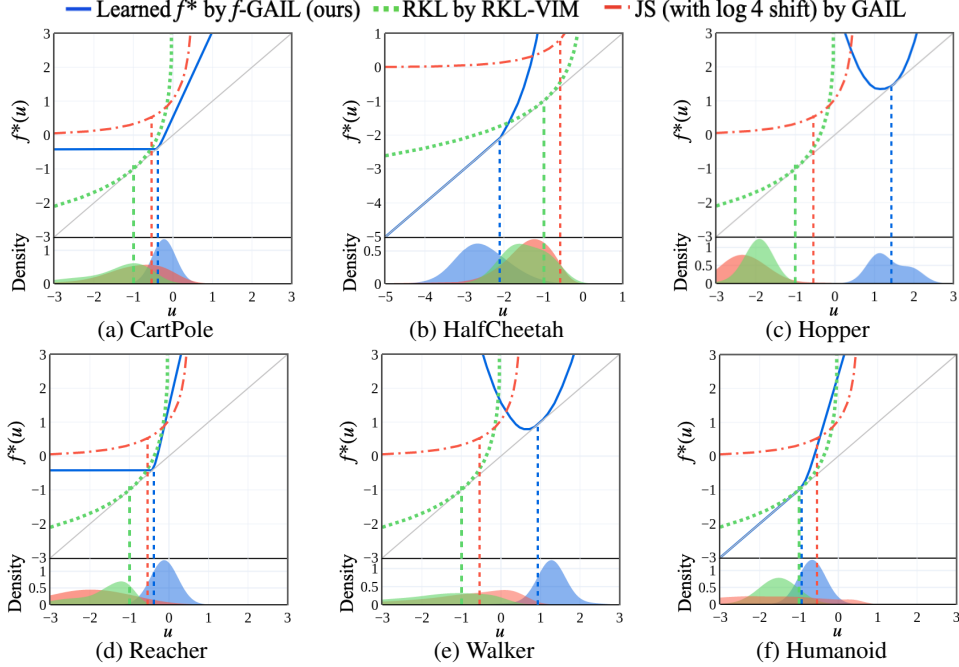

Figure 4: The learned $f_\phi^*(u)$ functions match the empirical input distributions at the zero gap regions with $f_\phi^*(u) - u \approx 0$, equivalently, $f(p(s,a)/q(s,a)) \approx f(1) = 0$, with close expert vs learner behavior distributions (i.e., $p$ vs $q$). The distributions of input $u$ were estimated by kernel density estimation [31] with Gaussian kernel of bandwidth $0.3$.

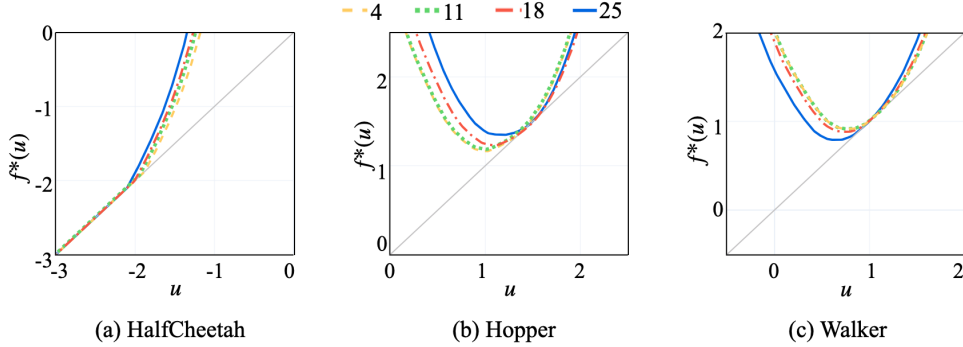

Figure 5: The learned $f_\phi^*(u)$ with different sample sizes.

In Fig. 4, we empirically estimated and showed the distributions of input $u$ for the state-of-the-art IL methods (including GAIL and RKL-VIM[5]) and our $f$-GAIL. Fig. 4 shows that overall $u$ distributions from our $f$-GAIL match the two criteria (i.e., close to zero gap and small standard deviation) better than baselines (See more statistical analysis on the two criteria across different approaches in Appendix B). This indicates that learner policies learned from $f$-GAIL are with smaller divergence, i.e., higher quality. We will provide experimental results on the learned policies to further validate this in Sec 4.2 below.

## 4.2 $f$-GAIL Performance in Policy Recovery

Fig. 6 shows the performances of our $f$-GAIL and all baselines under different training data sizes, and the tables in Appendix B provide detailed performance scores. In all tasks, our $f$-GAIL outperforms all the baselines. Especially, in more complex tasks, such as Hopper, Reacher, Walker, and Humanoid, $f$-GAIL shows a larger winning margin over the baselines, with at least $80\%$ of expert performances for all datasets. GAIL shows lower performances on complex tasks such as Hopper, Reacher, Walker, and Humanoid, comparing to simple tasks, i.e., CartPole and

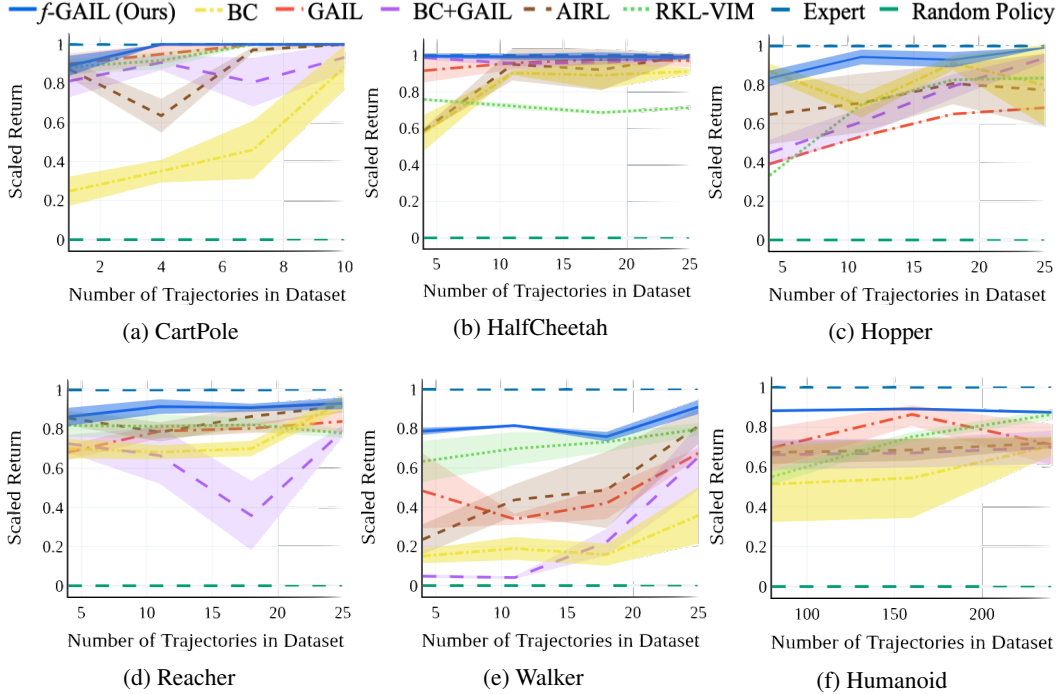

Figure 6: Performance of learned policies. The $y$-axis is the expected return (i.e., total reward), scaled so that the *expert* achieves 1 and a *random policy* achieves 0.

HalfCheetah (with much smaller state and action spaces). Overall, BC and BC initialized GAIL (BC+GAIL) have the lowest performances comparing to other baselines and our $f$-GAIL in all tasks. Moreover, they suffer from data efficiency problem, with extremely low performance when datasets are not sufficiently large. These results are consistent with that of [19], and the poor performances can be explained as a result of compounding error by covariate shift [27, 28].

AIRL performs poorly for Walker, with only 20% of expert performance when 4 trajectories were used for training, which increased up to 80% when using 25 trajectories. RKL-VIM had reasonable performances on CartPole, Hopper, Reacher, and Humanoid when sufficient amount of data was used, but was not able to get more than 80% expert performance for HalfCheetah, where our $f$-GAIL achieved expert performance. (See Tab. 6 in Appendix B for more detailed return values.) In terms of the convergence of the proposed $f$-GAIL, Fig. 7 below shows the training curve of $f$-divergence (i.e., the objective in eq. (5)) with respect to training epochs where it converges to less than $0.02$ for HalfCheetah after 450 epochs. Similar results were observed in other tasks.

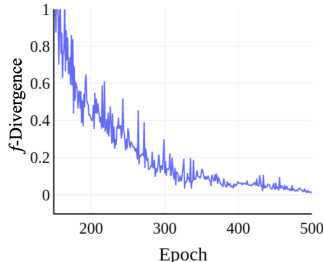

Figure 7: $f$-divergence curve in HalfCheetah.

## 4.3 Ablation Experiments

In this section, we investigate how structure choices of the proposed $f_\phi^*$ network, especially, the network expressiveness such as *the number of layers* and *the number of nodes per layer*, affect the model performance. In experiments, we took the CartPole, HalfCheetah and Reacher tasks as examples, and fixed the network structures of policy $\pi_\theta$ and the reward signal $T_\omega$. We changed the number of layers to be 1, 2, 4, and 7 (with 100 nodes each layer) and changed the number of nodes per layer to be 25, 50, 100 and 200 (with 4 layers). The comparison results are presented in Tab. 2. In simpler tasks with smaller state and action space, e.g. the CartPole, we observed quick convergence with $f$-GAIL, achieving expert return of 200. In this case, the structure choices do not have impact on the performance. However, in more complex tasks such as HalfCheetah and Reacher, a simple linear transformation of input (with one convex transformation layer) is not sufficient to learn a good policy function $\pi_\theta$. This naturally explains the better performances with the number of layers increased to 4 and the number of nodes per layer increased to 100. However, further increasing the number of layers to 7 and the number of nodes per layer to 200 decreased the performance a little bit. As a result, for

Table 2: Performances when changing *number of layers* and *number of nodes per layer* in $f_\phi^*$ network (Scores represent rewards. Higher scores indicate better learner policies).

| Task | Number of Layers (100 nodes per layer) | | | | Number of Nodes per Layer (4 layers) | | | |
|---|---|---|---|---|---|---|---|---|
| | 1 | 2 | 4 | 7 | 25 | 50 | 100 | 200 |
| HalfCheetah | 1539±144 | 4320±81 | **4445±79** | 4100±51 | 3546±132 | 4058±127 | **4445±79** | 4343±80 |
| Reacher | -22.8±4.2 | -16.4±3.2 | **-10.6±2.6** | -15.8±2.8 | -25.2±5.35 | -14.1±5.2 | **-10.6±2.6** | -12.6±4.0 |
| CartPole | 200±0 | | | | 200±0 | | | |

these tasks, 4 layers with each layer of 100 nodes suffice to represent an $f^*$-function. Consistent observations were made in other tasks, and we omit those results for brevity.

## 5 Discussion and Future Work

Our work makes the first attempt to model imitation learning with a learnable $f$-divergence from the underlying expert demonstrations. The model automatically learns an $f$-divergence between expert and learner behaviors, and a policy that produces expert-like behaviors.

**Meaning of the best $f$-divergence.** As a minimax optimization problem in eq. (5), $f$-GAIL searches for the best $f$-divergence in the "max" inner-loop *given the current learned policy $\pi$* learned from the "min" outer-loop, eventually leading to a stable solution of $(\pi, f^*)$. Here, given an expert demonstration dataset, a better divergence can measure the discrepancy more precisely than other divergences, thus enables training a learner with closer behaviors to the expert.

**Future work.** This work focuses on searching within the $f$-divergence space, where Wasserstein distance [17, 4] is not included. However, the divergence search space can be further extended to $c$-Wasserstein distance family [2], which subsumes $f$-divergence family and Wasserstein distance as special cases. Designing a network structure to represent $c$-Wasserstein distance family is challenging (we leave it as part of our future work), while a naive way is to model it as a convex combination of the $f$-divergence family (using our $f_\phi^*$ network) and Wasserstein distance. Moreover, beyond imitation learning, our $f^*$-network structure can be potentially "coupled" with $f$-GAN [25] and $f$-EBM [33] to learn an $f$-divergence between the generated vs real data distributions (e.g., image and audio files), which in turn trains a higher quality generator.

## Broader Impact

This paper aims to advance the imitation learning techniques, by learning an optimal discrepancy measure from $f$-divergence family, which has a wide range of applications in robotic engineering, system automation and control, etc. The authors do not expect the work will address or introduce any societal or ethical issues.

## Acknowledgments and Disclosure of Funding

Xin Zhang and Yanhua Li were supported in part by NSF grants IIS-1942680 (CAREER), CNS-1952085, CMMI-1831140, and DGE-2021871. Ziming Zhang was supported in part by NSF CCF-2006738. Zhi-Li Zhang was supported in part by NSF grants CMMI-1831140 and CNS-1901103.

## Footnotes

[3]Theoretically, given $\delta$ (defined as an infimum), it may not be achievable with a feasible $u \in \text{dom} f_\phi^*$. However, empirically, given the diverse input distributions of $f^*$ (See Sec 4.1), we can always introduce a projection operator [9] to limit the feasible space of $u$ for a better control of the shift operation. In our experiments, we never found any issue when directly applying Alg. 2 for the shifting operation.

[4]Both AIRL and RKL-VIM can be viewed as RKL divergence minimization problem. However, they use different lower bounds on RKL divergence (See details in [14] and [20, 25]).

[5]With AIRL, similar results were obtained as that of RKL-VIM, since they both employ RKL divergence (while using different lower bounds). We omitted the results for AIRL for brevity.

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
