[Supplementary Material]

# A Proof for Equation (7) in Section 3.2

In Section 3.2, we propose a shifting operation in eq. (7) to transform any convex function to a convex conjugate generator function of an $f$-divergence. Below, we summarize the shifting operation and prove its efficacy in proposition A.1.

**Proposition A.1.** *Given a convex function $f_\phi^* : dom_{f_\phi^*} \mapsto \mathbb{R}$, applying the shifting operation below transforms it to a convex conjugate generator function of an $f$-divergence,*

$$f_{\phi'}^*(u) = f_\phi^*(u - \frac{\delta}{2}) - \frac{\delta}{2}, \quad \text{where} \quad \delta = \inf_{u \in dom_{f_\phi^*}} \{f_\phi^*(u) - u\}. \tag{8}$$

*Proof.* As presented in Section 3.2, for an $f$-divergence, its convex conjugate generator function $f_{\phi'}^*(u)$ is *i)* convex, and *ii)* with zero gap from $u$, i.e., $\inf_{u \in dom_{f_{\phi'}^*}} \{f_{\phi'}^*(u) - u\} = 0$. Below, we prove that both these two constraints hold for the obtained $f_{\phi'}^*(u)$.

**Convexity.** Since a constant shift of a convex function preserves the convexity [9], the obtained $f_{\phi'}^*(u)$ is convex.

**Zero gap.** Given $\delta = \inf_{u \in \text{dom}_{\bar{f}^*}} \{\bar{f}^*(u) - u\}$, we denote the $\tilde{u}$ as the value that attains the infimum. Hence, we have $f_\phi^*(u) - u \geq \delta$ for $\forall u \in \text{dom}_{f_\phi^*}$. For the transformed function $f_{\phi'}^*(u) = f_\phi^*(u - \frac{\delta}{2}) - \frac{\delta}{2}$, we naturally have

$$f_{\phi'}^*(u) - u = f_\phi^*(u - \frac{\delta}{2}) - \frac{\delta}{2} - u = f_\phi^*(u - \frac{\delta}{2}) - (u - \frac{\delta}{2}) - \delta \geq \delta - \delta = 0, \quad \forall u \in \text{dom}_{f_\phi^*},$$

and the infimum is attained at $\tilde{u} + \frac{\delta}{2}$. This implies that the zero gap constraint $\inf_{u \in \text{dom}_{f_{\phi'}^*}} \{f_{\phi'}^*(u) - u\} = 0$ holds.

$\square$

# B Environments and Detailed Results

The environments we used for our experiments are from the OpenAI Gym [10] including the CartPole [8] from the classic RL literature, and five complex tasks simulated with MuJoCo [32], such as HalfCheetah, Hopper, Reacher, Walker, and Humanoid with task screenshots and version numbers shown in Fig. 8.

**Details of policy network structures.** The policy network structures $\pi_\theta$ of all the baselines and $f$-GAIL are the same in all experiments, with two hidden layers of 100 units each, and tanh nonlinearities in between. Note that behavior cloning (BC) employs the same structure to train a policy network with supervised learning.

**Details of reward signal network structures.** The reward signal network used in GAIL, BC+GAIL, AIRL, RKL-VIM and $f$-GAIL are all composed of three hidden layers of 100 units each with first two layers activated with tanh, and the final activation layers listed in Tab. 3.

**Details of $f_\phi^*$ network structure in $f$-GAIL.** For the study of the $f^*$ function in Sec 4.1 and the performances of the learned policy in Sec 4.2, the $f_\phi^*$ network is composed of 4 linear layers with hidden layer dimension of 100 and ReLU activation in between. For the ablation study in Sec 4.3, we changed the number of linear layers to be 1, 2, 4 and 7 (with 100 nodes per layer) and the number of nodes per layer to be 25, 50, 100, and 200 (with 4 layers).

**Evaluation setup.** For all the experiments, the amount of environment interaction used for GAIL, BC+GAIL, AIRL, RKL-VIM and the $f$-GAIL together with expert and random policy performances in each task is shown in Tab. 4. We followed GAIL [18] to fit value functions, with the same neural network architecture as the policy networks, and employed generalized advantage estimation [30] with $\gamma = 0.99$ and $\lambda = 0.95$, so that the gradient variance is reduced.

| (a) CartPole-v0 | (b) HalfCheetah-v2 | (c) Hopper-v2 |
| (d) Reacher-v2 | (e) Walker-v2 | (f) Humanoid-v2 |

Figure 8: Screenshots of six physics-based control tasks [32].

Table 3: Final layer activation functions for Reward Signal Networks.

| IL methods | Activation |
|---|---|
| GAIL | $\mathrm{Sigmoid}(v)$ |
| BC+GAIL | $\mathrm{Sigmoid}(v)$ |
| AIRL | $\mathrm{Sigmoid}(v)$ |
| RKL-VIM | $-\exp(v)$ |

Table 4: Parameters for baselines and $f$-GAIL.

| Task | Training iterations | Number of $(s, a)$ per iteration | Expert performance | Random policy performance |
|---|---|---|---|---|
| CartPole-v0 | 200 | 200 | 200±0 | 17± 4 |
| HalfCheetah-v2 | 500 | 2000 | 4501±118 | -901±49 |
| Hopper-v2 | 500 | 2000 | 3593±19 | 8± 6 |
| Reacher-v2 | 500 | 2000 | -4.5±1.7 | -93.7 ±4.8 |
| Walker-v2 | 500 | 2000 | 5657±33 | -2±3 |
| Humanoid-v2 | 700 | 30000 | 10400±55 | 101±26 |

## B.1 Detailed statistical results on Learned $f_\phi^*$ function

As explained in Sec 4.1, two criteria for the input distribution to the $f_\phi^*$ function govern the quality of the learned policy $\pi_\theta$, namely, *(i)* input $u$ centers around zero gap; *(ii)* input $u$ has small standard deviation. Now, based on Fig. 4, we analyze how much different IL methods satisfy the two criteria in all six tasks.

- To quantify criterion (i), we denote $\tilde{u}$ as the input value with zero gap, i.e., $f_\phi^*(\tilde{u}) - \tilde{u} = 0$, and $\bar{u}$ as the mean of the input $u$. Thus, we quantify the criterion (i) using the absolute difference between $\tilde{u}$ and $\bar{u}$, i.e., $\Delta_u = |\tilde{u} - \bar{u}|$.
- To quantify criterion (ii), we estimate the standard deviations $\sigma$ of input distributions for different IL methods in all tasks.

For both $\Delta_u$ and $\sigma$, the smaller values indicate a learner policy closer to expert policy. As a result, we examine their sum, i.e., $\Delta_u + \sigma$ as a unifying metric to evaluate overall how the two criteria are met. Tab. 5 shows the detailed results of $\Delta_u$, $\sigma$, and $\Delta_u + \sigma$. It shows that our proposed $f$-GAIL learns an $f_\phi^*$ function with consistently lower values on $\Delta_u + \sigma$, comparing to all baselines, which indicates that the learned $f_\phi^*$ function from $f$-GAIL can meet the two criteria better than baselines.

## B.2 Detailed results on learner policies

The exact learned policy return are listed in Tab. 6. The means and standard deviations are computed over 50 trajectories. A higher return indicates a better learned policy. All results are computed over 5 policies learned from random initializations.

Table 5: Analysis on input distributions of $f^*$ functions.

| Task | CartPole | HalfCheetah | Hopper | Reacher | Walker | Humanoid |
|---|---|---|---|---|---|---|
| $f$-GAIL | **0.28** | **0.62** | **0.51** | **0.60** | **0.49** | **0.52** |
| RKL-VIM | 1.25 | 0.96 | 1.36 | 2.14 | 4.62 | 2.85 |
| GAIL | 1.96 | 1.31 | 2.09 | 2.08 | 4.06 | 3.55 |

Table 6: Learned policy performance.

| Task | Datasize | BC | GAIL | BC+GAIL | AIRL | RKL-VIM | $f$-GAIL (Ours) |
|---|---|---|---|---|---|---|---|
| CartPole | 1 | 62±13 | **181±9** | 165±14 | 176±7 | 179±7 | 180±9 |
| | 4 | 81±10 | 191±9 | 183±7 | 133±15 | 185±8 | **200±0** |
| | 7 | 101±27 | **200±0** | 164±22 | 194±2 | **200±0** | **200±0** |
| | 10 | 178±20 | 199±0 | 187±13 | **200±0** | **200±0** | **200±0** |
| HalfCheetah | 4 | 2211±528 | 4047±344 | 4431±56 | 2276±65 | 3194±30 | **4481±60** |
| | 11 | 3979±61 | 4274±202 | 4263±90 | 4230±473 | 2994±94 | **4457±89** |
| | 18 | 3911±416 | 4377±135 | 4282±67 | 4073±605 | 2806±46 | **4461±132** |
| | 25 | 4027±91 | 4340±185 | 4447±48 | **4501±42** | 2952±45 | 4445±79 |
| Hopper | 4 | **3129±132** | 1413±26 | 1619±240 | 2328±549 | 1200±16 | 2996±142 |
| | 11 | 2491±218 | 1923±16 | 2188±257 | 2539±544 | 2513±3 | **3390±135** |
| | 18 | 3276±133 | 2336±10 | 2849±224 | 2898±362 | 2969±17 | **3339±142** |
| | 25 | 2868±745 | 2452±12 | 3372±79 | 2779±675 | 3001±42 | **3561±6** |
| Reacher | 4 | -31.3±4.4 | -33.0± 3.5 | -29.0±4.0 | -17.4±3.3 | -20.7±5.2 | **-16.7±4.0** |
| | 11 | -32.9±3.1 | -23.4± 3.2 | -34.4±12.8 | -23.7±4.3 | -21.1±5.4 | **-12.1±3.3** |
| | 18 | -31.3±3.4 | -22.1± 2.1 | -61.8±15.7 | -16.6±4.4 | -20.4±3.1 | **-12.6±1.8** |
| | 25 | **-10.0±3.2** | -18.9± 5.0 | -23.2±2.4 | -11.8±2.9 | -24.2±2.0 | -10.6±2.6 |
| Walker2d | 4 | 848±206 | 2728±1079 | 267±50 | 1327±431 | 3577±594 | **4448±103** |
| | 11 | 1068±328 | 1911±160 | 226±36 | 2466±454 | 3947±475 | **4609±22** |
| | 18 | 888±316 | 2372±453 | 1251±378 | 2755±1103 | 4138±287 | **4290±139** |
| | 25 | 2018±812 | 3816±148 | 3700±939 | 4599±504 | 4507±179 | **5148±205** |
| Humanoid | 80 | 5391±3918 | 7268±2101 | 6908±1577 | 7034±591 | 5772±409 | **9180±49** |
| | 160 | 5713±4126 | 8994±1053 | 7003±1488 | 7160±559 | 7842±245 | **9280±68** |
| | 240 | 7378±998 | 7430±2106 | 7294±1705 | 7528±273 | 8993±252 | **9130±114** |