[Reviews · NeurIPS 2020]

Review 1

Summary and Contributions: The paper proposes an imitation learning algorithm that as opposed to the existing variations of generative adversarial imitation learning (GAIL) algorithms, instead of using a pre-defined f-divergences such as JS, KL, RKL, etc. learns an f-divergence. They model the convex conjugate of f with a neural network that meets the constraints that f-divergence requires. The algorithm closely follows GAIL with the difference that an f-divergence is learned along with the reward function.

Strengths: The paper is well-written and it's easy to follow. It is working on the important problem of imitation learning which is very relevant to the NeurIPS community. It addresses an issue in the recent state of the art adversarial imitation learning algorithms which use specific f-divergence functions. The proposed method unifies these algorithms and proposes a simple general algorithm that learns the divergence function instead. Each of the pre-known f-divergence functions has their own limitations, for instance one is mode-covering and one is mode-seeking. Learning the f-divergence function helps the algorithm find the function that is suitable for that specific problem. Also the results show that for similar problems similar functions are learned. The algorithm is tested in performance against some pre-known f-divergence functions in some MuJoCo tasks and the results show improvement compared to baselines in all cases.

Weaknesses: In terms of novelty, the algorithm is a small modification of previous adversarial imitation learning algorithms. Testing the algorithm on more complex domains would have been better for evaluation of the capabilities of the algorithm.

Correctness: The method seems to be sound and the empirical method is correct.

Clarity: The paper is well-written and it's easy to follow.

Relation to Prior Work: The paper has properly cited and clearly discussed the relation to previous work.

Reproducibility: Yes

Additional Feedback:


Review 2

Summary and Contributions: The paper proposes to generalise GAIL to the f-divergence formulation, with a fully learnable discriminator that can approximate arbitrary f-divergence. Extra care is taken to ensure that the numerical model satisfy the constraints of f-divergence. Empirical results also support the proposed method.

Strengths: The proposed method is clean and sound, and implements the necessary care to ensure that the model approximates f-divergence. The empirical results are thorough and detailed, demonstrating benefits of the proposed method.

Weaknesses: I have several concerns about the proposed method: 1) while it may be implicit from the main text, a rigorous proof to show that zero-gap and convex constraints are both necessary and sufficient conditions for learning f-divergence should be presented. 2) While the empirical section covers many important details about the actual implementation, it is common that the implementations of adversarial imitation learning contains subtle details that impact the evaluation. For instance, how many random seeds are used to obtain the results? How were the standard deviations obtained for each task? 3) It seems that all baselines are implemented under the model f*(T(s, a)). For known divergences, a primary advantage would be the ability to simplify the objective function (e.g. the original GAIL objective). Comparison against these optimised versions of baselines methods are necessary for fairness.

Correctness: I have several concerns about the two criteria proposed in Section 4.1. 1) how are the input distributions for each method computed (e.g. from (s,a) sampled from the learned policies, or expert (s,a))? 2) it is unclear that the two criteria could be used to assess the quality of the learned policy. The current claim seems to rely on the assumption that that f*(T(s, a)) is a good discriminator between the expert and learner, which may not be necessarily true. Perhaps the authors should consider using expert's (s,a) to first show that f*(T(s, a)) could achieve a divergence value of near-zero, before analysing the learner policy.

Clarity: The paper is clearly written and easy to follow.

Relation to Prior Work: There is sufficient details about previous work and how they are related to the proposed method.

Reproducibility: No

Additional Feedback: After rebuttal: I thank the authors for their responses. My concerns with sufficient and necessary conditions, and divergence evaluations are addressed. My concerns over the implementation details, and the performance comparison against is baselines remain relatively open. For instance, the performances on baselines seem low compared to reported results from the original papers (e.g. walker, hopper on GAIL). I agree with other reviewers' assessment that more evaluation could further support the method. As such, I would keep my current score.


Review 3

Summary and Contributions: The authors propose a novel imitation learning approach. The approach tries to find the best f-divergence to minimize and then minimizes it with an adversarial objective. To do so, the authors learn an explicit representation of the fenchel conjugate, using a special network architecture as well as a projection approach to satisfy the necessary constraints.

Strengths: The idea of finding the best divergence to match the expert’s occupancy measure is novel and could lead to improved performance in imitation learning algorithms.

Weaknesses: The proposed formulation is not completely well-defined and the evaluation does not show what the authors claim.

Correctness: One main concern is the choice of experiments which is highly problematic when evaluating the algorithms capability to match the distribution of the expert. In all domains except Reacher, it is sufficient to match a specific and mostly constant feature. For walkers, this feature is the forward velocity while for CartPole this is the angle. The Reacher domain is only slightly more complex as the agent has to match the difference between the given target and the x-y position; however, here too the agent only has to match a constant value rather than a distribution. As the paper is about divergences, it is problematic that the experiments do not include a task where the agent has to match a specific distribution rather than reproduce a constant value to achieve a high score. Another concern is the metric used in the main evaluation. The authors compare the sample-efficiency of different approaches and use this to claim superiority of the new approach. At no point in the paper do the authors elaborate how different divergences may affect the sample-complexity in imitation learning. Furthermore, the authors compare against methods that estimate the divergence in different ways. It is very plausible that the differences are better explained by the ability to accurately estimate any divergence from a limited number of samples rather than by the divergence itself.

Clarity: Overall, the paper is well written and easy to understand.

Relation to Prior Work: The authors cover the most relevant related works and the idea is novel.

Reproducibility: Yes

Additional Feedback: My other main concern is that the objective in Eq. (5) is badly motivated and the implications are under underexplored. The imitation learning objective is notoriously ill-defined and a large part of the literature focuses on introducing objectives that produce good behavior. The notion of finding the “best” f-divergence therefore requires us to state what we are optimizing for, which the authors don’t do very explicitly. On line 38, the authors mention that an imitation learning method which uses a fixed divergence method is likely to learn a sub-optimal policy, but the notion of optimality does not exist without a given divergence. For example, whether mode-seeking or mode-covering behavior is better is entirely dependent on context that the agent does not have. Either solution could be better. The authors mention that the best divergence should be the largest one, but this notion is not elaborated on and the implications are unclear. Figure 4 suggests that the agent does converge to a specific f-divergence. It would be good if the meaning of the divergence could be explored further. This is especially true for the divergences shown for Hopper and Walker2d whose conjugate looks very different from the RKL and JS conjugates. Furthermore, it would be good to show the variance on the learned conjugates, i.e. does the algorithm always converge to a similar divergence? Another point: While it is correct that BC minimizes the KL divergence on the policy, it is misleading to group it with the other methods as the other methods minimize the divergence on the state-action joint distributions. The latter requires an understanding of the environment dynamics and can therefore be much more effective. A minor point: since states are generally continuous in this work, P should map to [0, inf) on line 62


Review 4

Summary and Contributions: The paper presents an approach for learning an f-divergence measure as part of the discriminator training process within a GAIL framework for imitation learning. In particular, the discriminator function is divided into a reward signal function and an f-divergence function. Several necessary conditions are imposed onto the f-divergence function architecture to satisfy the requirements for it to be an admissible f-divergence measure. The f-divergence is trained jointly with the reward signal function as a part of discriminator training. The learned f-divergence is shown to improve performance over pre-defined divergence measures, such as JS, KL and RKL divergences, in multiple OpenAI Gym tasks.

Strengths: - Imposing additional structure on the discriminator while at the same time keeping it differentiable is a very interesting way to add inductive bias and improve performance. - The proposed method is shown to significantly improve performance on the evaluated tasks over pre-defined divergence measures.

Weaknesses: In the presented approach, training of the f-divergence function is based on the discriminator loss and it is not directly influenced by the performance of the learned policy. Although we can see improvements of the final policy performance, it would be interesting to see a discussion whether optimizing discriminator loss would always lead to an increased policy performance and if using the policy performance as an additional training objective for the f-measure would further improve the performance (e.g. in a meta-learning setting).

Correctness: Claims and mathematical derivations in the paper are coherent and empirical evaluation correctly shows performance improvements over the baselines.

Clarity: The paper is well-written and easy to understand and follow.

Relation to Prior Work: The paper clearly establishes connection to prior works and uses them as baselines for the experiments.

Reproducibility: Yes

Additional Feedback: Post-rebuttal comments: Thanks for addressing reviewer's comments and providing new details in the rebuttal. As mentioned by other reviewers, the notion of optimality of f-divergence is somewhat ambiguous and currently mostly supported by the empirical evidence, so it would be great to see more clarification regarding this in the final version of the paper.

[Author Response · NeurIPS 2020]

Thank reviewers for the comments. Please find our responses below, *with reference indices consistent with the paper.*

**To reviewer 3. Q3-1: Experimental choices.** Imitation learning (IL)
aims to match the state-action distributions between the learner and the
expert, rather than the goal feature(s). The expert state-action distributions
from the data are high-dimensional and stochastic. Taking CartPole as
an example, Fig. 1 (right) shows the expert (stochastic) policy on two
sample states, where the states are from a 4-dimensional continuous space
(see Appendix B in [18]). Moreover, Fig. 1 (left) shows that in the expert
demonstrations, the angle (feature) is NOT simply a constant (over 250
trajectories in CartPole). In addition, for fair comparisons, our evaluation

Figure 1: Angle dynamics (left) and policy (right) in CartPole from expert data.

tasks/settings are consistent with SOTA [13,18,20]. **Q3-2: Sample efficiency.** Results on sample efficiency are
presented in Fig. 5 in the paper, where $f$-GAIL outperforms baselines over different sample sizes (Sec 4.2). Moreover,
it is true that $f$-GAIL can more accurately estimate $f$-divergence from a limited number of samples. Fig. 2 below shows
that given a task, the learned $f^*$ functions are consistent with different sample sizes. In fact, the choice of $f$-divergence
matters. The better divergence estimation accuracy enables $f$-GAIL to examine and compare $f$-divergence choices,
which is why $f$-GAIL consistently outperforms baselines.

**Q3-3: Meaning of the "best" $f$-divergence.** Our $f$-
GAIL is defined as a minimax optimization problem in
eq.(5) in the paper. The best $f$-divergence is searched
in the "max" inner-loop *given the current learned policy*
$\pi$ learned from the "min" outer-loop, eventually leading
to a stable solution of $(\pi, f^*)$. **Q3-4: The optimality**
**depends on the divergence and context?** The notion of
optimality exists and depends on the expert demonstration
data, rather than the divergence, namely, the optimality

Figure 2: The learned $f_\phi^*(u)$ with different sample sizes

refers to the smallest discrepancy of behavior distributions (in state-action pairs) between the learner and the expert.
Given an expert demonstration dataset, a better divergence can measure the discrepancy more precisely than other
divergences, thus enable training a learner with closer behaviors to the expert. In the example of whether mode-seeking
or -covering makes sense, it depends on the expert demonstration data (NOT the context), i.e., whether the expert was
performing mode-seeking or -covering when generating the data. **Q3-5: Meaning of the learned divergence? The**
**variance of the learned divergence?** It is nontrivial to find an analytical close-form function to express the learned
$f^*$, due to the huge convex function space with $f(1) = 0$. We leave this as our future work. Fig. 2 above shows that
given a task, the learned $f^*$ functions are consistent (small variance) for different sample sizes. **Q3-6: Comparison**
**with BC.** We agree that BC minimizes the policy KL divergence as what we noted in Sec. 4 (line 200). We included
BC as a baseline for completeness, namely, a comprehensive comparison with SOTA. **Q3-7: Notation of $\mathcal{P}$ in line 62.**
Our notation represents a probability distribution of transitioning from $(s, a)$ to a next state $s'$, thus the outcome is in
$[0, 1]$. It is consistent with the literature, e.g., Sec. 2 in [Yu et al. arXiv:1909.09314].

Table 1: Baseline performances in HalfCheetah.

| Datasize | GAIL | GAIL$_f$ |
|---|---|---|
| 4 | 4047±344 | 4055±257 |
| 25 | 4340±185 | 4472±166 |

**To reviewer 2. Q2-1: Necessity and sufficiency of two constraints.** The $f$-
divergence definition requires the generator $f$ function to be convex and $f(1) = 0$
[11,23,24]. Convex and zero-gap constraints are necessary and sufficient conditions to
guarantee an $f$-divergence, based on $f^{**} = f$ (see §3.3.2 in [9]) for convex functions,
i.e., $f(1) = f^{**}(1) = \max_u\{u - f^*(u)\} = 0$. **Q2-2: Implementation details.** We
used 5 random seeds with mean and variance calculated over 50 trajectories (see Sec. 4 and Appendix B). These
settings are consistent with SOTA [11,23,24]. **Q2-3: Baselines.** All baselines were implemented with their original
models [13,18,20], rather than $f^*(T(s, a))$. In fact, as shown in Tab. 1 (with GAIL as the original model and GAIL$_f$
as $f^*(T(s, a))$), the baseline results are similar, when implemented using original models vs $f^*(T(s, a))$. **Q2-4:** The
input state distributions were sampled from expert demonstrations. **Q2-5: Divergence evaluation.** Following your
suggestion, Fig. 3 below shows the training curve of $f$-divergence wrt. epochs where it converges to less than 0.02 for
HalfCheetah after 450 epochs. Similar results were observed in other tasks.

**To reviewer 1. Q1-1: Novelty:** We are the *first* to model imitation learning with a learnable
$f$-divergence measure (using the proposed $f^*$-network), rather than a predefined divergence,
which yields better learner policies than the literature on GAIL [13,18,20]. **Q1-2: More**
**complex tasks:** We evaluated $f$-GAIL on tasks consistent with SOTA [13,18,20], including
Humanoid, with the high state dimension of 376. We plan to evaluate $f$-GAIL on more
complex tasks, e.g., Simitate [Memmesheimer et al. arXiv:1905.06002].

**To reviewer 5. Training objective:** In our Alg 1, all three networks are trained with the
same objective in eq.(5), using *adversarial training*. The updating gradients ($\nabla_\omega$ and $\nabla_\phi$)
are obtained by taking the derivative of eq.(5) wrt. $\omega$ and $\phi$, respectively, while fixing $\pi_\theta$.

Figure 3: $f$-divergence curve in HalfCheetah.

The objective for policy $\pi_\theta$ is the same as eq.(5) with $T_\omega$ and $f_\phi^*$ fixed. We will add such details in the final paper.

[Meta-Review · NeurIPS 2020]

After reading the authors' rebuttal, the reviewers discussed their concerns about this paper. Ultimately, a consensus was not reached as reviewer #3 feels that some of her/his concerns were not properly addressed in the authors' feedback. The other reviewers are positive with respect to the paper (especially thanks to the promising experimental results), but they share one of the concerns of reviewer #3, i.e., the definition of ``optimal f-divergence'' and the convergence properties of the proposed approach. I agree with them that the paper has merits and the ideas contained in the paper are interesting, so I propose to accept it, but I recommend that the authors take the issues raised in the reviews seriously and address them carefully in the final version of the paper.